# Medication Non-Adherence in Inflammatory Bowel Disease: A Systematic Review Identifying Risk Factors and Opportunities for Intervention

**DOI:** 10.3390/pharmacy13010021

**Published:** 2025-02-07

**Authors:** Kathryn King, Wladyslawa Czuber-Dochan, Trudie Chalder, Christine Norton

**Affiliations:** 1Florence Nightingale Faculty of Nursing Midwifery and Palliative Care, King’s College London, London SE1 8WA, UK; wladzia.czuber-dochan@kcl.ac.uk (W.C.-D.); christine.norton@kcl.ac.uk (C.N.); 2Department of Psychological Medicine, Institute of Psychiatry, Psychology and Neuroscience, King’s College London, London SE5 8AB, UK; trudie.chalder@kcl.ac.uk

**Keywords:** inflammatory bowel disease, medication non-adherence, medication non-concordance, medication non-compliance, medication non-persistence, systematic review

## Abstract

Inflammatory bowel disease (IBD) is treated with medications to induce and maintain remission. However, many people with IBD do not take their prescribed treatment. Identifying factors associated with IBD medication adherence is crucial for supporting effective disease management and maintaining remission. Quantitative and qualitative studies researching IBD medication adherence between 2011 and 2023 were reviewed. In total, 36,589 participants were included in 79 studies. The associated non-adherence factors were contradictory across studies, with rates notably higher (72–79%) when measured via medication refill. Non-adherence was lower in high-quality studies using self-report measures (10.7–28.7%). The frequent modifiable non-adherence risks were a poor understanding of treatment or disease, medication accessibility and an individual’s organisation and planning. Clinical variables relating to non-adherence were the treatment type, drug regime and disease activity. Depression, negative treatment beliefs/mood and anxiety increased the non-adherence likelihood. The non-modifiable factors of limited finance, younger age and female sex were also risks. Side effects were the main reason cited for IBD non-adherence in interviews. A large, contradictory set of literature exists regarding the factors underpinning IBD non-adherence, influenced by the adherence measures used. Simpler medication regimes and improved accessibility would help to improve adherence. IBD education could enhance patient knowledge and beliefs. Reminders and cues might minimise forgetting medication. Modifying risks through an adherence support intervention could improve outcomes.

## 1. Introduction

Inflammatory bowel disease (IBD) refers to the chronic inflammatory diseases of Crohn’s disease (CD) and ulcerative colitis (UC). Worldwide, there were 4.90 million IBD cases in 2019 [1], an almost 50% increase since the 1990s. Historically, IBD has been most prevalent in developed regions; however, recently there has been a rapid rise in incidence within the Middle East, Asia, and South America.

These incurable conditions are associated with an excessive immune response leading to unpredictable disease course, impacting quality of life and causing long-term consequences such as gut damage and colorectal cancer [2]. The most frequently reported symptoms in remission are fatigue, chronic pain, incontinence and extra-intestinal manifestations such as arthritis. Diagnostic examinations are typically reported as “painful” and “stressful” by patients [3]. Medication therapy aims for “tight control” of inflammatory activity and to induce and maintain symptomatic, endoscopic and histological remission whilst reducing the risk of sequalae [4]. Some medications also decrease the incidence of colorectal cancer, e.g., mesalazine. However, the protective effect requires strict adherence. High non-adherence prevalence (up to 72%) has been reported across a range of IBD drugs and healthcare systems [5]. Non-adherence to IBD medications can significantly impact treatment outcomes, with studies associating it with increased risks of disease flare [6] and a reduced quality of life [7]. Non-adherence in IBD leads to high healthcare and societal costs [8,9,10].

Researching and understanding adherence is complex. Defining, measuring and identifying patients with a high possibility of non-adherence, as well as understanding and supporting medication adherence, is a challenge [11]. A combination of determinants have been found to influence non-adherence, including patient-related and healthcare-related factors [12]. Yet typically, studies have investigated only a one or two of these individually.

This is mirrored throughout healthcare, across multiple health conditions. Several theories have been proposed as to why people are non-adherent to their medication, with recognition that some factors can be modifiable and can be addressed. This includes health psychology and cognitive behavioural theories [13] (health belief model, social cognitive theory, and theory of planned behaviour) which consider an individual’s cognition as a key behaviour change factor. Alternatively, biopsychosocial models attribute various physical and psychosocial influencers to explain non-adherence [14].

Knowing why a person with IBD is non-adherent would enable the development of tailored and effective interventions to improve self-management and adherence in this chronic condition, whilst reducing costs. Previous work has considered either adherence, non-adherence or related concepts in IBD individually or focussed upon specific medication types [12,15]. Earlier scoping and systematic reviews identified the complexity of related factors on non-adherence across a range of conditions [16,17], yet a comprehensive systematic review focussed upon IBD is lacking. This piece will build upon previous reviews of all IBD medication non-adherence terms over an extended time period. The aim of this review is to systematically explore and synthesise the available evidence of both modifiable and non-modifiable factors associated with non-adherence in people with IBD. This will help to identify both modifiable targets for health interventions to enhance and maintain adherence and non-modifiable targets which should be clinically monitored and supported wherever possible to minimise non-adherence.

## 2. Materials and Methods


**Search strategy**


Six electronic databases were searched systematically in November 2023. Published articles from peer-reviewed journals relevant to the review’s aims were identified. The reference lists of the included studies were searched for appropriate papers. A combination of terms relating to adherence and IBD was used to search the databases. A full list of search terms, adapted for each database, is presented in Appendix A. Retrieved studies were exported into EndNote (Version 20) and transferred to the Covidence (Version 2) reference management software. Bias was minimised through two reviewers (K.K. and C.N.) screening 50% each of the titles and abstracts of retrieved papers for eligibility, according to pre-determined inclusion criteria. Reviewers were assigned full-text papers for data extraction, with K.K. performing double data extraction. Any disagreements were resolved through discussion with a third reviewer (W.C.D.). A flow diagram (Appendix A) reports the study selection process and provides reasons for inclusion and exclusion as suggested by the PRISMA-P guidelines [18]. The protocol was registered in PROSPERO [CRD42021240056].


**Inclusion Criteria**


All papers in English, published from 2011 to November 2023, where the majority of participants were ≥16 years old, with a diagnosis of IBD and prescribed one or more medication for IBD, were included. A cut-off of 12 years was considered extensive for the previous literature, whilst considering contemporary, relevant IBD medications. Papers were excluded if the study population were all children/young people (<16 years old), not living with IBD or not prescribed medication for IBD. Bias was minimised through conducting a thorough review of available published literature. Peer-reviewed papers of qualitative or quantitative study design were included investigating factors associated with adherence and non-adherence in adults living with IBD. Grey literature was not considered due to the volume of papers identified within the specified period. Intervention studies, reviews/protocols or conference abstracts were excluded from this review, as were papers not written in English.

As the review’s aim was to investigate factors associated with any type of non-adherence, where papers reported adherence, outcomes were reversed to non-adherence to ensure meaningful comparisons. When studies used alternative terms to describe not taking medication as prescribed (compliance/non-compliance, concordance/non-concordance and persistence/non-persistence), these were also included. When studies differentiated between persistence/non-persistence or discontinuation and non-adherence, the study was only included if non-adherence/adherence was reported separately as a primary outcome (See Appendix A).


**Analyses**


Most studies included were quantitative and did not control for potential confounders, presenting only univariate (one variable) or bivariate (two variables) data analysis. Due to the need to control for other factors within a model, the increased likelihood of larger samples being used in multivariable analysis (MVA) and the large number of studies found overall, only factors significant in MVA were considered most likely related to non-adherence. If methods were stated as multiple linear regression, multiple logistic regression, multivariate analysis of variance (MANOVA), factor analysis, cluster analysis or multivariable analysis, we included this as MVA. If data were stated as being statistically significant at univariate or bivariate analysis, but non-significant at MVA, this was also reported.


**Quality Appraisal**


The Critical Appraisal Skills Programme (CASP) tools were used to assess the quality of both qualitative and quantitative papers. Qualitative studies were appraised using the CASP checklist for qualitative data [19]. For quantitative papers, in line with the CASP tool recommendations [20], a CASP scoring system was not used and a systematic rating system was devised for quality rating by the research team. Each quantitative study was given a total base score of “three”; one point was subtracted if the study did not use a reliable, recognised adherence measure and one point was subtracted if authors did not specify the use of a form of MVA. This resulted in scores of three (high), two (medium) or one (low). If study reporting was unclear and/or with limited data, the study was reviewed again by reviewers and scores amended. No studies were excluded based on quality.

## 3. Results

A total of 7596 papers were identified from six databases and the reference lists of the included studies. After screening titles and abstracts, 384 papers remained with full-text eligibility screening. A total of 79 papers were identified for the review, undergoing data extraction. Studies were conducted between 2011 and 2023, across the world, including Europe (32 studies), North America (20 studies), South America (4 studies) Asia (18 studies), and Oceania (3 studies). Two studies conducted their research multi-nationally [8,21]. While many included single sites (38 studies) based within general or tertiary hospitals settings, some were multi-site (32 studies) or not site-specific (9 studies, e.g., online).


**Demographics of participants**


In total, 36,589 participants were included, ranging from 7 to 6048 per study (Appendix A). Ages ranged from 15 to 81 years, although not all studies reported this clearly. Most studies had both male and female participants, except three which had 100% female participants [22,23,24]. The race or ethnicity of participants was reported by one study only [25].

Most studies did not present smoking or alcohol use and for those which did, participants were largely non-smokers.

Most participants were in full-time employment. Education levels were mixed. When relationship status was reported, participants were mainly married, in a relationship/currently partnered and/or living together with their partner.

Studies categorised the disease type as either UC, CD, IBD, IBD unclassified (IBDU), indeterminate ulcerative colitis (IUC) and unknown, with 2 not reporting the disease type (2.53%) and 34 investigating both CD and UC (43%). A total of 4 studies (5.1%) investigated exclusively CD, whereas 19 (24.1%) focussed on UC. Eighteen studies also categorised IUC or IBDU (22.8%). Two studies did not distinguish between IBD types [8,26].

Time since diagnosis was frequently reported, ranging from 0.1 years to 51 years. A variety of medication classes, routes, regimes and doses were presented, with almost a third included all medication types (27 studies, 34.2%), and 11 (13.9%) not stating this.


**Study design**


Sixty-six studies (83.5%) were quantitative and ten (12.7%) used mixed methods [9,27,28,29,30,31,32,33,34,35]. The remaining three (3.8%) were qualitative [36,37,38], with data analysis following grounded theory principles to develop themes and associated links in one study [37]. Forty-six studies (58%) were cross-sectional design, through online or face to face questionnaires at a single centre. The remaining study designs were either prospective (10), longitudinal (2), retrospective (12), observational (5), cohort (1) or interviews/focus groups (3). The study length was from 1 month to 13 years. Fifteen studies did not report the length of data collection (see Appendix A).

The *p* values considered as significant were typically <0.05, with either univariate or a range of multivariate analyses conducted.

A theoretical framework was used by nine studies (11.4%) to explain adherence, their choice of an adherence measure or their findings. 

Quality appraisal rated most studies as medium in quality (36) or high (35), and eight were of low quality. Several studies presented unclear reporting of their results [11,22,27,39,40,41]. 


**Measuring and Categorisation of Non-adherence**


The 79 studies used a wide variety of definitions and tools to measure non-adherence. Consequently, a huge range from 4.3% to 88.9% in non-adherence is presented in Figure 1, alongside alternative classifications (when a study used multiple tools for measuring adherence, an overall non-adherence value was calculated). Very few studies found non-adherence to be under 20%.

Cut-offs for defining adherence/non-adherence were dependent upon the adherence measure used. The most popular cut-off was 80% adherence, whereby non-adherence was taken as the use of equal to [42] or less than 80% of the prescribed medication [3,43,44]. Alternatively, good adherence was defined as taking >80% of the prescribed doses [44]. Similarly, if the medication possession ratio (MPR) was ≥80% for an aminosalicylate (5-ASA) treatment, this was frequently rated as good adherence [2].

Sub-group analysis of high-quality, multi-centre studies with over 100 participants showed distinct differences between non-adherence rates through validated self-report questionnaires (10.7–49.8%) and medication possession ratios (72–79%).

Typically, 21–30% of participants were non-adherent to their medication. Ten (12.7%) papers did not report any non-adherence/adherence rates or anything similar [28,36,41,45,46,47,48,49,50] or were vague in their categorisation, such as “not good adherence” (7%) [51] or “partial non-adherence” at 20% [52] or 18% [26]. Some authors were consistent in their use of terms, such as “low”, “medium” or “high” non-adherence/adherence, yet studies varied in their definition of these terms, sometimes with minimal or no definition. For example, “low adherence” referred to both 3% [52] and 49.8% [53] in different studies, whereas “inadequate” adherence in one study included all participants answering “rarely,” “sometimes,” “often” or “always” when asked “How often do you miss medication intake”? [29]. 

The classification of taking or not taking medication as prescribed was most commonly referred to as “adherence” or “non-adherence” (see Appendix A). However, not taking medication as prescribed was occasionally defined as “poor adherence” [21] or “low adherence” [39,52,53,54,55], amongst other descriptors. 

Table 1 shows the measures of non-adherence reported. Typically, quantitative tools were used. The Morisky medication adherence scale (MMAS) was utilised in 26 studies and author-designed, non-validated questionnaires in 17. Scores on the MMAS often ranged between 4 and 6 out of 8 and were considered as a self-report of “good” adherence. Scoring for validated measures was in line with the recommended guidelines. For example, scoring 4–16 out of a possible 20 in the four studies using the medication adherence scale.

Report Scale-4 (MARS-4) [6,45,57,58] was used to assess non-adherence to medication [56]. The MARS-5 [40,48,59,60] and 10-item scale [52] were also used, as well as one MARS scale unspecified [56]. Additional measures included monitoring of therapeutic drug levels [25], medication possession ratio in 14 studies (MPR; percentage of prescribed medication dispensed to a patient during a specific period/over a period of refill intervals) [23,56] and persistence evaluated over one year after an index prescription [2]. Pill counts over varying time periods [42,82], e.g., two months [42], were used in two studies. Sub-group analysis of high-quality studies showed non-adherence ranged from 21.7 to 49.8% when using the validated MMAS measure, whereas this ranged from 10.7 to 28.7% with the MARS measure. Qualitative studies also used a variety of tools to elicit medication concerns [37]. 


**Strongest and most consistent associations with non-adherence**


Knowledge and understanding of IBD and its treatment had the strongest and most consistent associations with non-adherence, with 92% in the reviewed quantitative studies being significant. Accessibility, organisation and planning were positively correlated with significant results in 80% of investigations using quantitative studies. Qualitative studies also emphasised the impact of forgetting, poor medication availability and disorganisation as the main modifiable non-adherence causes.

Modifiable treatment-related factors (such as treatment type, route and regimens) were frequently discussed in quantitative and qualitative studies; most were positively associated with non-adherence.

Modifiable psychological factors were also significantly positively associated with non-adherence in 72% of investigations.

Several non-modifiable patient demographics were reported in quantitative studies. Most significant was living in poor residential areas, associated with a reduced life quality and socioeconomic status [8,20,25,32,45,48,81]. Finance, medication and increased care cost difficulties were found to be frequently associated with non-adherence risks [7,25,32,34,44,67,77,85], along with the demographics of age, disease activity and sex.


**Factors associated with non-adherence**


Within categories associated with non-adherence, specific non-modifiable and modifiable factors were identified throughout the literature. The findings were often contradictory, with minimal agreement, and will be discussed in greater detail.

Unless otherwise stated, the findings presented are significant under MVA, presented in categories and as individual factors in Table 2 and summarised in the text. The non-adherence risk generally increased with the greater number of significant risk factors experienced [80].


**Demographics (non-modifiable):**



**Age**


Twenty-four studies reported on age with MVA. There were 15 positive associations with age and non-adherence, 3 mixed (positively and/or negatively associated and non-significant age group dependent), and only 6 non-significant associations in different studies. Participants within the ages of 15–29 years were most likely to be non-adherent, while those ≥61 years had greater likelihood of being adherent. Similarly, being below 60 years was also found to be associated with lower adherence compared to over 60 years [2]. 


**Age at diagnosis**


Four studies explored age at diagnosis in relation to adherence, with mixed findings. Two studies [74,84] found that people being diagnosed at a younger age (up to 29 years), were more likely to non-adhere. Yet, two other studies [25,72] reported non-significant results.


**Sex**


Eleven studies analysed the associations of sex with non-adherence. Females were more likely to be non-adherent in seven studies [2,11,53,59,62,83,90]. One study reported only females with UC showing higher non-adherence [59], whereas another found this for CD [83]. Females under 40 years old [53] had greatest risk for non-adherence. In contrast, males were significantly more adherent [2]. However, contradictory findings were also presented in two studies [53,67] with non-adherence higher in males.


**Race**


Race was only reported in one study [25]. Non-adherence was most common in participants of African–Caribbean descent, although this was not significant. 


**Diagnosis (non-modifiable):**


Twenty-two studies reported mixed results regarding diagnosis. 


**Disease type**


Eleven studies explored the type of IBD. Four found CD to be associated with non-adherence [54,60,89,91], yet five reported this as non-significant [6,8,33,75,93]. Three studies found UC was not a significant predictor of non-adherence [8,75,93].


**Disease activity**


Eleven studies investigated disease activity. Participants with active disease were more likely to be non-adherent in three studies [74,86,93]. The relationship between highly active CD and non-adherence through avoiding infusions was associated with pain, diarrhoea or being admitted as an inpatient and receiving alternative treatment [93]. A negative relationship was reported between active disease and non-adherence in one study [33], yet only in patients experiencing at least one relapse in the past 12 months. 

In other studies, participants in remission [6,57], those with a lower probability of relapse [66] or an absence of abdominal symptoms (such as visible bleeding) [72] were most likely to be non-adherent. Yet, six studies found these relationships or being in pain due to IBD to be non-significant [11,25,26,66,72,74].


**Disease duration**


Participants with a “long” diagnosis duration of between 6 and 15 years were reported to be more adherent than those with a shorter diagnosis in a single study [33]. Contradicting this, three studies [32,57,83] found non-adherence increased with time since diagnosis. Two studies investigated a “short” diagnosis duration of less than 5 years with non-adherence, one finding a significant relationship [78] and the other a non-significant one [74].


**General health (modifiable/non-modifiable)**


General health was reported by four studies [29,55,66,84], with three modifiable and non-modifiable factors related to non-adherence. The most frequent general health risk factor for IBD non-adherence was taking treatment for another chronic condition [29,66], with one study finding this to only be significant when the IBD medication was 5-ASA. Having comorbidities was also associated with non-adherence in IBD [66].

Conversely, another study found that individuals not prescribed other chronic treatment were at an up to 2.2 times higher risk of non-adherence with their IBD medication than those who were [84].


**Treatment (modifiable)**


Treatment, including medication type and mode, route, dose, regimen frequency, convenience of administration and adverse effects, was the most investigated modifiable factor, analysed by MVA in 28 studies with 35 positive relationships with non-adherence.


**Drug-Class**


Frequently associated with non-adherence was being prescribed either aminosalicylates [11,30,31,70,71,78] or biologics [6,73,75,87,88]. Mesalamine was a significant predictor of non-adherence, compared to other drugs [70]. One study reported a non-significant relationship between non-adherence and aminosalicylates [8].

Patients who had never switched from an index aminosalicylate were much more likely to be non-adherent than those who changed (*p* < 0.0001), with up to 76.9% non-adherence [90]. Patients with no history of switching from any drug type from their index medication were also likely to be non-adherent [90].

Within different mesalamine types, oral Pentasa had the lowest adherence rate (26.4%), whereas adherence to Mezavant once daily was significantly higher (40.9%) than other oral treatments (*p* < 0.001). Only one study reported lower adherence in patients prescribed non-immunomodulators (*p* = 0.049) than aminosalicylates [3]. In this study, despite aminosalicylates having the highest non-adherence rates, this was not significant. In the same study [3], biologics were related to adherent behaviours, supporting other research [57].

Although mixed significant results were also found for biologic medications, if prescribed either biologic/combination biological–immunomodulator therapy, this was the only factor associated with low adherence when starting on anti-TNFs in one study [6] and the single factor correlated with non-adherence in another [75]. Regardless of whether patients were treated with biologics intravenously or subcutaneously [73,75,87,88], this medication increased the non-adherence risk.

Four studies found that the prescribed medication type did not have an impact upon non-adherence, whether these were 5-ASAs, biologics, steroids or immunosuppressants [8,11,32,89] (*p* < 0.05).


**Route**


Eleven studies investigated the route of administration, with this most frequently relating to higher non-adherence. An uncomfortable medication route, i.e., subcutaneous as opposed to oral treatments [28] or via infusion, was associated with not taking it as prescribed [6,88]. When patients had never used rectal 5-ASAs, this increased their risk of not taking oral 5-ASA medication [90].


**Frequency/Regime**


Four studies reported the relationship between non-adherence and frequent/multiple medications [33,66,79,82] or long-term treatments [62,82]. This included initiating treatment on multiple daily dosing of either balsalazide, mesalamine-delayed release (Asacol) or sulfalazine [90]. A regimen of 40 mg adalimumab every other week was a predictor for missing medication, as opposed to 40 mg weekly. However, increasing adalimumab to 80 mg every other week was a predictor of improved adherence.

Pill burden (the effort of taking all prescribed drug/s) was a risk for non-adherence in one study [47], yet it was also reported as non-significant [30]. In contrast, when patients were prescribed fewer than eight daily tablets [44], non-adherence was increasingly likely. Yet monotherapy was found to have a non-significant effect on non-adherence [75].


**Side effects**


Side effects and non-adherence were reported only by three studies, with a positive relationship in two [30,56] and non-significant in one [66].


**Future non-adherence**


Current non-adherence was found to be an independent predictor for future non-adherence [74].


**Healthcare (modifiable)**


Healthcare was frequently investigated for its relationship with non-adherence (17 studies), with 18 modifiable risk factors identified [7,9,11,27,30,40,65,66,72,82,83,86,90]. Non-adherence was most likely when patients experienced negative relations and/or poor communication with healthcare providers [9,11,30,72], if no specialist or tailored care was received [27,40,90] or if frequent inpatient hospitalisation or emergency care was experienced [7,86]. Frequent IBD outpatient [65,82] and general health appointments and adherence monitoring were also associated with adherence promotion [7]. The risk of non-adherence significantly increased if a patient received minimal treatment information from their team [66] or the importance of medication adherence was not reinforced [7].

When patients found contacting their gastroenterologist easy, adherence improved [29].


**Habits (modifiable)**


Modifiable habits were investigated by 11 studies, the most common being smoking—in 10. Four studies found current smoking to be a non-adherence risk factor [3,23,29,54]. In one study, smoking was one of a few significant factors [3]. Smoking highly influenced non-adherence in specific cohort—in patients prescribed thiopurines [29] or oral 5-ASA [23]. Male smokers also showed a significant relationship with non-adherence (*p* = 0.018) [23] whereas females did not, with these all being non-smokers. Similarly, another study found non-smoking participants to be more adherent [54]. Yet, smoking was not significantly associated with not taking medication in five studies [73,74,87,89,93].

Alcohol consumption was investigated by two studies, with mixed findings. Consumption (but not frequency) was related to low adherence (*p* = 0.029) in males [23]. This was not significant in females [23] nor patients overall [58]. Prescribed narcotic use was only explored by one study [89], with non-significant findings.


**Diet (modifiable)**


Modifiable dietary factors were reported by four studies. Regularly eating alone [71] or missing a meal [79] and not storing medications where meals are eaten [42] were all positive predictors of non-adherence. The use of nutritional supplements protected against non-adherence [31].


**Finance (modifiable/non-modifiable)**


Eight studies reported that a mixture of modifiable and non-modifiable financial variables increased non-adherence, specifically having public or non-commercial insurance as opposed to private [88,89,90], lower socioeconomic status [25] or overall higher healthcare costs (including inpatient, outpatient and emergency) [7].


**Living location (non-modifiable)**


Six studies identified living location as a predictor for non-adherence [8,26,41,63,66,90]. Residing in a socially deprived area and/or reduced life quality significantly predicted non-adherence in all studies that reported this [8,26,41,63,66,90].

Country and areas of residence however had mixed results. Living in north-east, south or west America was positively associated with taking medication [90], whereas living in the mid-west was not significant [90]. Residing in the UK over Australia was negatively associated with non-adherence [8].


**Employment (non-modifiable)**


Five studies investigated employment. Demanding, busy work constraints relating to treatment increased non-adherence [56,73]. Full-time employment and/or a greater number of working hours were also positively associated with non-adherence [11].

“Permanent employment”, however, was not significantly correlated with non-adherence [30], and “self-employment” reduced the risk of non-adherence [70].


**Education (non-modifiable)**


Five studies reported on education level and its impact upon taking medication [32,58,59,70,78]. Those with a combined higher socioeconomic status, occupational and educational level were more non-adherent, but they were not significant as individual factors in the same study [32].


**Social support and relationships (modifiable/non-modifiable)**


Five studies investigated social support and relationships [30,57,59,70,93]. Non-adherence was associated with being single [70] or receiving poor social support, whether emotional or tangible [30] or low informational social support [57]. Having to “deal with friends” when living with CD in fact increased the risk of non-adherence, but not in UC [59].


**Psychological factors (modifiable)**


Psychological health was one of the most frequently modifiable factors investigated, but with contradictory results. Twenty-five studies were carried out, with treatment beliefs, perceptions and concerns explored in seventeen studies. Negative beliefs, mediated by poor patient satisfaction, led to low adherence [40]. When treatment was thought ineffective [30,66,74] or unnecessary by the patient [8,56,69,71], this was also positively related to non-adherence (seven studies). Conversely, when individuals expressed a need for their IBD treatment, this was protective against non-adherence [58]. Disease beliefs were also significantly related to non-adherence—specifically when IBD was perceived as having a short illness duration [6,59,64], with one study finding significance only for participants with CD [59]. A weaker illness identity (fewer IBD associated symptoms) was also a risk for low adherence [6], although the illness perceptions of the daily consequences of living with IBD were not [6].

Treatment concerns, namely side [47,69] and adverse effects [58], predicted low medication adherence (in three studies), regardless of medication type. However, this was non-significant in four studies [6,8,66,71].

When participants expressed a reduced sense of control over their IBD [11,74], lower perceived competence with treatment regime [72] or experienced religious or spiritual struggle using negative coping strategies [46], non-adherence increased.

Nine studies researched depression and poor psychological states, seven of which identified these as significant non-adherence risks [25,30,58,66,67,74,92].

Anxiety was a significant risk for non-adherence in all four studies investigating this factor [30,58,73,74]. When UC had less of an impact on an individual’s mood [74] or the patient was indifferent or sceptical regarding treatment benefit [56], these were largely predictive of non-adherence. Experiencing a stronger, negative emotional response to having IBD [6] was also associated with not taking medication as prescribed. Only one study found mood did not influence adherence [73].


**Accessibility, organisation and planning (modifiable)**


Ten studies reported on personal awareness and planning [9,30,40,42,44,47,63,67,71,82]. Missing scheduled appointments [30,71], having a lower priority for medications [47], not being as careful when taking medications [9] and medication doses at weekends [82] were found to be significant non-adherence predictors. Not using tools such as dosette boxes or cues to action, e.g., alarms or reminders to take medication, whether from family or healthcare providers, was also a risk for non-adherence [42,44]. Mixed evidence was found for forgetting or carelessness and not keeping medications accessible when due, reporting both significantly positive [40,42] and non-significant [63,67] relationships with not taking medication.


**Knowledge and understanding (modifiable)**


This was investigated by 12 studies, with 10 showing positive associations with non-adherence [11,28,31,41,58,59,65,66,73,74]. Having poor disease or treatment knowledge [11,58,59,65,74] or limited medical information recall [28] was related to non-adherence. Yet one study found this only in UC patients [59] and another only for knowledge of azathioprine [58]. These factors were also related with non-adherence when participants reported poor communication with [39] or inadequate medication information from healthcare professionals [66].

Not being keen on using the internet [41] was a significant risk for non-adherence, whereas having “high curiosity levels” [30] was not.

Patient organisation membership increased adherence in one study [8] and increased non-adherence in another [73].


**Alternative treatment (modifiable)**


Of two studies investigating alternative and complementary therapies, one found patients seeking holistic health approaches to be more likely to reduce prescribed IBD medication, as compared to those who did not (30% vs. 16%, *p* = 0.02) [63]. Yet individuals using complementary therapies for general health showed similar non-adherence to those using prescribed medications. Complementary therapy use for IBD was also non-significant [75].


**Specific cohorts (non-modifiable)**


Cohort specific factors investigated included pregnant and post-partum women [22,23,24,38], with no studies specifically analysing non-adherence risk at the MVA level. However, non-adherence led to a significantly increased likelihood for disease relapse and adverse pregnancy outcomes at the MVA level, particularly in women taking oral mesalamine [24].

Two studies researched non-adherence throughout the COVID-19 pandemic; again, there were none at the MVA level [52]. The greatest non-adherence reason was fear of attending hospital (50% participants), due to the perception of catching infections because of their IBD, being immunosuppressed [77] or having higher medication concerns [52].


**Qualitative studies**


Thirteen studies presented qualitative data, three were purely qualitative [36,37,38], and nine predominantly quantitative with free text comments [9,27,28,29,30,31,32,33,34,35] (Table 3).

Reported reasons for non-adherence, are presented in themes (Table 4), with direct quotations (Table 5).


**Treatment**


The most frequent type of treatment adherence barriers expressed were adverse effects [29,30,32,33,34,36,37,38]. Uncertainty regarding drug safety was also common [38].

The administration mode [30,32,34,36,37], including rectal [34], and self-administered subcutaneous injections, was reported as a challenge [36]. Several drugs taken multiple times a day were considered to pose treatment difficulties, resulting in “pill fatigue” [37]. A desire for a “simpler regimen” [36] was often expressed.


**Finance**


Repeated prescription costs were also reported as a non-adherence reason [29,30,32,36,37].


**Lifestyle**


Travelling and/or being in public spaces [34] with busy lifestyles were problematic for adhering. This included carrying medications to work [32] or difficulties renewing prescriptions when away from home [32,33,36,37].


**Beliefs**


Personal opinions regarding treatment necessity often influenced medication discontinuation, altering doses and missing intermittent ones [29,30,33,34,36,37,38]. Reasons also included non-acceptance of diagnosis, leading to a negative emotional response towards treatment [36].


**Forgetting and Organisation**


Forgetting was a main reason for not taking medications [29,30,32,33,36,37]. Timing with due treatment or general disorganisation were also adherence barriers [37].


**Accessibility**


Accessing medications was a common adherence challenge [29,32,33,36,37], as was repeatedly refilling medications [36,37].


**Pregnancy and pregnancy planning**


Being pregnant and avoiding harm to the baby was a frequent concern of women using a range of medications [33,38]. Women also spoke about safety and uncertainty of teratogenic effects if currently pregnant or planning pregnancy, in addition to fertility concerns from females and males [38].

## 4. Discussion

This is the most recent comprehensive international review that outlines the complexities and challenges of non-adherence to prescribed medication in IBD today. Between 4.3% to 88.9% of patients were identified to be non-adherent, with at least 30% in most studies and a lack of consensus on defining adherence/non-adherence [28,29,35,87,93]. The large range of adherence cut-offs, depending upon the instrument’s purpose [44,86], maximised differences between adherent and non-adherent IBD patients, potentially leading to inaccurate measurement of these concepts. Multiple versions of the same tool to measure adherence made study comparisons difficult.

Individual studies suggest adherence rates differ due to a range of complex, modifiable and non-modifiable factors that could be intentional and/or unintentional [87]. Interestingly, the majority of reviewed studies did not measure “intentional” or “unintentional” concepts or explore the reasoning behind non-adherence. Consequently, a large, inconsistent, often poorly reported and contradictory set of literature exists, making it challenging to draw specific clinical conclusions from this review.

Knowledge and understanding of IBD and its treatment were the most frequent modifiable predictors of non-adherence, with 92% of associations in the reviewed quantitative studies being significant for non-adherence. Low disease knowledge can be influenced by diagnosis uncertainty [29,53], leading to classification bias for patients and research. This review included CD, UC and indeterminate colitis and their range of treatments, which were not wholly comparable. Agreed validated international case definitions for IBD are required to clarity patient understanding whilst minimising the risk of misclassification, impact upon non-adherence and data misinterpretation [23].

Lack of understanding of IBD and medication benefits [8,11,58,59,65,74], common in newly diagnosed patients, can significantly impact non-adherence. Improved patient understanding of the disease and the need for continuous medication requires clear, concise education regarding IBD and its treatment, provided by multidisciplinary teams [56]. If not delivered effectively by specialist clinicians [96], IBD patients may recall only 50% of information from appointments [28]. This can lead to poor adherence soon after the first consultation [48], meaning the reinforcement of key medication messages is critical. Also vital at the treatment recommendation stage is determining the likelihood of patients taking medication. This will allow the clinician to work through together with the patient to target any challenges or barriers. Socratic questioning to elicit personal circumstances together with Motivational Interviewing techniques such as “How likely are you to take your medication?” and “What would help you in taking it?” have been shown to be particularly effective [97].

The impact of self-education is inconsistent [98], as this review shows it may not facilitate adherence [31]. Patients have a desire for self-teaching surrounding their long-term condition and medication-related knowledge [41]. Internet use is popular to support active learning, promote disease understanding and evaluate medical advice, but has the potential to be inaccurate and/or misleading and may not fully meet patients’ expectations, leading to poorer adherence [99]. Furthermore, adherence research using the internet may exclude those without access. Alternatively, when healthcare professionals take time to provide accurate guidance, patients can feel more confident about managing their IBD [72], reducing their anxiety and encouraging more timely follow-ups [100]. Yet, clinicians must remain mindful of knowledge and understanding developing within the same individual through experience, potentially impacting upon adherence changes [47]. This demands a need for personalised educational interventions, rather than generic solutions [43], for adherence promotion [41].

Accessibility, organisation and planning were positively correlated with significant results in 80% of investigations by quantitative studies. Qualitative studies emphasised the impact of forgetting, poor medication availability and disorganisation as the main modifiable non-adherence causes. Lack of routine, busy lifestyles, including full-time employment [11,56,73] and medication regimes interfering with daily activities commonly increase the likelihood of forgetting, leading to non-adherence [29,43,79,82], particularly if lower treatment priority is expressed. Forgetting can be effectively modified with strategies such as setting alarms and placing medication close to traditional reminders (e.g., toothbrush or kettle) [47,56,67,78,101]. Reminders and feedback from healthcare professionals can also be an effective, inexpensive method to enhance clinical practice and medication use [12,17,102]. Memory cues help to prevent a diminished sense of treatment priority [44]. Medication dispensers or pill cases are reportedly strongest at predicting good adherence [42], easily determining whether medication has been taken. Adherence interventions containing such technical components have demonstrated consistent benefits over time [103]. Reward approaches have also shown improvements when combined with these technical strategies, although further research is recommended [104].

Modifiable treatment-related factors were frequently discussed in quantitative and qualitative studies, most were positively associated with non-adherence. IBD has the complexity of multiple medications, supplements and variable regimes, which can significantly impact non-adherence—found in some studies to be almost 90% [105]. Medication side effects [28,29,105] also significantly determine adherence, potentially increasing the use of “complementary and alternative medicine.”

IBD treatment administered via various routes [37] may cause discomfort (e.g., injections, rectal medications or oral tablets that are difficult/large to swallow) and may be associated with greater non-adherence risks [47,86], impacting life quality [43]. Yet despite this, some non-adherence data collection tools are designed solely for one medication route and/or type [28]. Furthermore, techniques such as MPR have been identified to have variable calculation methods, significantly affecting adherence estimates [106]. Thus, the validity and verifiability of study results should be interpreted with caution. A variety of methods collecting a combination of adherence barriers and disease activity have been suggested, although data interpretation gathered from a range of approaches can be challenging, increasing the potential for disagreement.

Non-adherence to oral 5-ASA was frequent in this review [11,30,31,70,71,78], correlating with previous reports [51,65] ranging from 38 to 60% [107]. Prescribed for less severe, more stable disease, this requires minimal monitoring, often leading to greater non-adherence [90]. Patients typically identify quiescent periods with recovery, with a reduced need for treatment [47,56,58,76]. Aminosalicylates are also associated with decreased frequency and dosing, with monotherapy reporting higher non-adherence rates, as opposed to combination therapy [67]. Monitoring adherence using targeted strategies in stable patients is therefore necessary [44].

Other IBD medications taken infrequently (e.g., immunosuppressants) necessitate specific storage and/or require attendance to hospital appointments for administration, raising non-adherence risks [36]. Conversely, multiple inconvenient dosing regimens influence developing routines and habits [43,44,64,86]. Similarly, previous literature reported 30% of patients when asked reasons for non-adherence, answered “*too many pills*” [10]. Despite a long-established inverse relationship between the dosing regimen complexity and non-adherence in IBD [101,108], complicated “three times daily” dosing regimens are still used by many gastroenterologists [64]. To achieve better clinical IBD outcomes, the findings from this review and prior evidence support simplifying daily regimes wherever possible [2,56,68,79].

Patient preferences must be identified in both clinical prescribing and reversed supervision (prospective prescriptions considering individual retrospective medication adherence) [26,43]. For research, specific aims investigating single medication regimes in one chronic disease is encouraged, to more accurately identify non-adherence predictors [53].

Modifiable psychological factors were significantly positively associated with non-adherence in 72% of investigations, supporting previous systematic reviews [16,17]. Depression was the most common, followed closely by anxiety, then patients less bothered about the treatment benefit or their IBD having a lower impact on mood. Depression is a risk in many chronic disease populations [25,26,109,110], frequently associated with stress, “feeling hassled” and significant life events, together possibly contributing to non-adherence and IBD relapse [53,111]. Concerns regarding medication safety and adverse effects regarding long-term maintenance medication [3,10,107] cause further treatment stressors and “adherence barriers” [72].

Purposefully not taking prescribed medication can often be the main reason for intentional non-adherence. Ranging from 70.7 to 97%, it is typically higher than non-intentional non-adherence, associated with treatment doubts when feeling well and/or not experiencing effective action, particularly if an individual considers that the treatment disadvantages outweigh the benefits [44]. This perception of the necessity for medication can be impacted by prior experiences, with the importance of necessity reducing over time [69].

Yet voluntary non-adherence is more challenging to address. These non-adherence difficulties may not be disclosed to healthcare professionals [75], particularly if negative relations exist involving poor communication [9,11,30,72] and a lack of specialist healthcare [27,90]. This review and previous studies and reviews found such modifiable healthcare factors positively associated with non-adherence [16]. To promote good patient–physician relations, reinforcing empathy and leniency is recommended [75]. This helps by avoiding putting patients in a defensive state when asking them to self-report adherence behaviours, achieving honest, reliable answers [75]. Despite this, it is argued that most people report the truth when questioned about their adherence [25,26]. Many studies conclude that the combined use of self-reporting along with a more validated, objective adherence measurement is appropriate for a greater understanding of non-adherence reasons [25,35,40,42,43,48,77,78,81,82].

Once non-adherence is identified, intervention strategies must actively involve those patients choosing not to take IBD medication as prescribed [78], promoting their awareness of the non-adherence consequences [23,24,53]. When individuals view their medications more positively, they are more likely to adhere to it [30,40]. However, this tailored approach in specialist healthcare with overstretched services could be challenging. To ensure that the care needs of people with IBD are focussed upon [41], a balanced approach with multidisciplinary teams supporting patients to access offline/online resources [81,109], offering accurate, comprehensive and holistic IBD education, will subsequently help to promote knowledge and self-management [53,74].

Several non-modifiable patient demographics were reported in quantitative studies. Most significant was living in poor residential areas, associated with reduced life quality and socioeconomic status [8,25,26,41,63,66,90]. Finance, medication and increased care cost difficulties also indirectly impact psychological stability and adherence [7,25,26,29,30,32,37,86], referred to as “downstream non-adherence consequences” [86], rather than being direct predictors [41]. Further research is warranted to explore these complex contributory factors to non-adherence [30,78].

Age, disease activity and sex were non-modifiable demographics associated with significant non-adherence risks. Yet age and disease activity were also not significant in some studies, supporting the previous inconsistent literature [73]. Younger people do not necessarily prioritise medication use, focussing upon leisure, going out and friendships [23] as opposed to discussing health conditions and concerns with others [107]. The contradictory results for these factors could be attributed to the high levels of heterogeneity in the studies compared. Most specifically, methodological weaknesses were brought about by many smaller, single-site, retrospective studies with lower sample sizes. Additional sub-group analysis of specific data highlighted the limitations of comparing multiple adherence measurements across a range of studies, namely self-report and medication refills. As secondary measures of adherence, medication refills are a popular, relatively straightforward method for avoiding the subjective bias of inaccurate patient recall. However, they have been known to inaccurately estimate adherence and it is impossible to determine whether a patient has accurately taken their medication [112]. The PDC is considered a more accurate, suitable method, focusing on days the patient is “covered” or supplied with medication [113].

Simple, universal interventions for these non-modifiable factors reflect similar outcomes, often producing non-significant improvements in non-adherers [69]. Special care should be taken to increase medication adherence in youngsters with IBD [81], particularly with IBD incidence in adolescents increasing [26]. Younger patients need support to modify their treatment, thought processes on adherence and non-adherence consequences [46]. Simpler drug delivery regimes whilst monitoring patients are beneficial [53], combined with supervised, smooth care transitions to adult services [26].

Specifically, the non-adherence in young females with IBD identified in this review [2,47,53,55,90] is consistent with previous findings [110]. Social embarrassment of IBD and enema use are suggested reasons [27,111]. As IBD patients are largely affected during child-bearing age, females frequently express treatment concerns specific to reproductive journeys, fertility, pregnancy and lactation fears, leading to non-adherence [38,47,114], verified by qualitative data. Previous research shows pregnant women often overestimate the potential harm of their IBD medication, with many of those choosing to breastfeed discontinuing treatment (74%) [115]. Enhancing the quality and quantity of accurate, accessible reproductive health and IBD information available for patients is necessary, as opposed to potentially seeking limited, non-evidence-based information online [38]. Timely, bespoke reproductive counselling from gastroenterologists, reinforcing importance of adherence before, during and after pregnancy is effective [114]. This close working of clinician and patient in a supported, communicative manner bridges vital information gaps in reproductive health and IBD whilst reducing flare-ups and modifying non-adherence [38].


**Strengths and Limitations**


The literature for this review spans 12 years, identified from a broad range of extensive databases representing medical, nursing, health, psychology and scientific disciplines, from clinical, academic and research data. With healthcare changing rapidly, variability is huge across the identified studies. The selected period allows a consideration of the unprecedented, life-changing experiences (e.g., COVID-19 pandemic, cost of living crisis) [77] which critically impacted medication use and data collection. However, these times may also limit accuracy and generalisability. Considering such prior practice may be incompatible, opposing current patient care, particularly following the pandemic or immunomodulator use [86].

This international literature review considered a huge variety of factors including healthcare, cultures, insurance, prescribing and medications, clinic and medication accessibility, availability and disease-related knowledge [2,21,29,38,53,76], enhancing generalisability. These must be considered when investigating non-adherence in patients.

Yet, cross-continental comparison of the same medication type incorporates national drug variations due to formulations, prescribing practices, treatment availability, diverse patient-funding of prescriptions and biased patient profiles [21,27,50]. Additional difficulties arise when validated tools are translated into alternative languages, with assorted interpretations. A range of adherence and non-adherence terms being used synonymously across the literature adds further interpretative complexity. Non-adherence rates may also be determined by the adherence measure/s used, which may not be wholly comparable.

The review inclusion criteria were pre-defined but generous, with no design restrictions, resulting in a large quantity of studies, varied patient cohorts and study designs. However, this led to challenges in synthesising data. Overall, a large sample [2,26,29,38,114] more accurately represents the adherence of general populations in real-world clinical settings as opposed to clinical trials. Yet this is limited by minimal demographic data collection [38]. Individually, many studies had the strengths of focussing upon “select cohorts,” particularly those who were non-adherent [53], from single centres [9,22,25,26,34,35,37,39,45,46,47,51,54,55,58,59,61,62,63,64,65,66,67,68,69,70,71,75,77,78,81,83,84,87,88,89,91,92], typically outpatient or tertiary [3,8,22,24,25,26,32,35,42,43,44,46,47,51,52,54,56,58,59,62,63,66,67,69,70,77,78,83,84,87,89,92], with small samples [3,24,28,34,35,40,57,68,81,82,85]. However, such literature may not generate “meaningful” results [26,57]. Future research acknowledges a need for replication with larger, more diverse samples from multi-centres [3,42,81] and extended follow-ups, enhancing representation whilst increasing validity [58].

58%of the reviewed studies were cross-sectional in design [8,9,11,21,22,23,25,27,28,30,31,32,39,40,41,42,44,45,46,47,49,50,51,52,53,54,55,56,57,58,59,61,62,64,66,68,70,71,72,75,76,77,78,79,81,92], typically at a single timepoint, over short periods, meaning it was impossible to evaluate the suggested strategies for adherence promotion [42]. Longitudinal studies have challenges regarding pharmacy medication records maintaining accuracy and consistency. Studies presenting retrospective research [2,23,26,44] are limited to previous behaviours. To predict future non-adherence and evaluate strategies for adherence promotion, further prospective research is necessary [26,42,82].

A specific study limitation identified from this review and earlier research [114], was the restricted inclusion criteria, with some studies only including participants of a certain diagnosis length [41,50]. This may minimise the influence of disease duration, shown to impact IBD non-adherence [32,57,78,83]. Using only papers published in English and with methodological differences of selection [2,42,44] restricts the representativeness, generalisability and persuasiveness to community practice.

Biases also arose from the investigation of specific adherence-related factors [81], with some studies failing to report on known risks (e.g., smoking, body mass index, employment or socioeconomic status) [35,53,54] or the reasoning for non-adherence [24].

The analysis of this review was thorough and detailed, considering both qualitative and quantitative studies. Focussing on significance at a multivariate level ensured confounding factors were eliminated, whilst identifying those most likely related to non-adherence.

Finally, a main limitating challenge within all adherence data is the true prevalence rate [26]. Typically, non-adherers are the least likely to participate in research or attend clinics [57,69], potentially masking and underrepresenting their perspectives, trends and behaviours. Moving forward, utilising accurate prescription databases with more clinical data collection may overcome this, comparing responders with non-responders [114].

## 5. Conclusions

Treatment adherence is a critical component in maintaining remission, alongside other clinical and biological factors. This review has identified many modifiable and non-modifiable factors having mixed relationships with non-adherence in IBD, thus offering an improved understanding of determinants of adherence and non-adherence.

In future practice, multidisciplinary clinicians must collaborate with patients throughout their IBD journey. Firstly, identifying barriers and challenges patients foresee regarding taking their IBD medication through active listening and questioning. Clinicians aware of non-modifiable factors can better identify patients at risk of non-adherence and develop targeted strategies to support them. Problem solving targeting modifiable adherence barriers could reverse and modify active patient decisions of not taking treatment.

Unrealistic modifiable medication fears must be addressed through education to enable clear knowledge and understanding of IBD and treatment. Healthcare professionals should enhance patients’ self-management strategies, offering accurate resources for independent learning. Various technical and reward strategies could be suggested to patients to improve their organisation and planning of treatment taking. Ongoing patient monitoring of the psychological and physical impact of IBD with personalised adherence support is required. A “one-size-fits-all” approach must be avoided, as the underlying causes and common barriers may differ considerably, necessitating varied interventions.

For future research, a unified, formalised definition of non-adherence is urgently needed, with consideration of how theoretical models of adherence could inform future research. This will help to further clarify between intentional and non-intentional non-adherence and modifiable and non-modifiable factors. It is critical to utilise a range of measures to help to determine objective, accurate non-adherence rates. Additional qualitative investigations will also identify reasoning behind non-adherent behaviours.

Further investigation of adherence promotion interventions tailored to the most salient non-adherence risk factors including knowledge and understanding of IBD and treatment, accessibility, organisation and planning, forgetting, poor medication availability, treatment-related factors (type, route and regimens) and modifiable psychological factors is also critical. Specifically, further research to minimise forgetting and regarding the impact of reward strategies is warranted.

Informed development and implementation of adherence support programmes will ultimately improve individual health outcomes, quality of life and health-related costs.

## Figures and Tables

**Figure 1 pharmacy-13-00021-f001:**
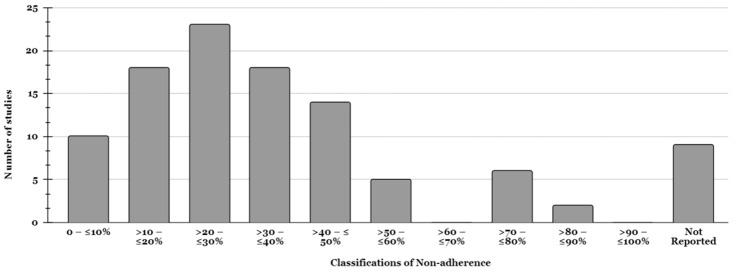
Rates of non-adherence. (Note: The range of non-adherence (4.3–88.9%) reflects variations in study populations, adherence definitions and measurement methodologies).

**Table 1 pharmacy-13-00021-t001:** Quantitative measures of non-adherence.

Measures of Non-Adherence	Version	Studies
Self-report: medication adherence report scale (MARS)	Not specified	[56]
	4	[6,45,57,58]
	5	[40,48,59,60]
	10-item scale	[52]
Self-report: Morisky medication adherence scale (MMAS)	Non-specific (4 items)	[33,49,50,61,62,63,64,65]
	Non-specific (6 item)	[21]
	Non-specific (8 item)	[6,9,39,41,54,55,66,67,68,69]
	For IBD patients (8 item)	[11,25,46,53,70,71,72]
Self-report: Medication adherence (non-validated)	(23 items)	[3]
Self-report: Visual analogue scale (VAS)	Not reported	[24,73,74]
Self-report: QUOTE-IBD questionnaire	Non-standardised multiple-choice test with open-ended questions	[27]
Self-report: study’s own questionnaire	E.g. individual questions, Likert scale, ordinal scale	[9,22,28,29,31,32,41,42,43,47,51,69,75,76,77,78,79,80]
Self-report: verbal	Amount/dose of treatment or incidents taken or missed within a specific time period	[33,42,44,81,82,83]
Medical records	Reviewed by researchers	[42,44,77]
Pill count	(Short-term measure of adherence)	[42,82]
Medication possession ratio (MPR) *	% of prescribed medication dispensed to a patient during a specific period	[2,7,23,34,35,69,71,84,85,86,87,88,89,90]
Proportion of days covered (PDC)	Number of any oral 5-ASA drug on hand during a 1-year period (different to MPR).	[90]
Blood tests (thiopurine levels)		[25,26]
Other		[34,60,91,92,93]
None reported		[36]

Key: * The medication possession ratio (MPR) is the proportion of medication supply dispensed, presuming that the previous prescription was not filled within the first and last dispensed date [94,95]. Adherence using MPR is usually defined as ≥80%, and non-adherence <80%.

**Table 2 pharmacy-13-00021-t002:** Factors associated with non-adherence/low adherence in studies using multivariate analysis (mva), multiple logistic regression or factor analysis.

Categories	Individual Factors Associated with Non-Adherence/Low Adherence	Relationship	Studies
Age	Younger age (15–29 years)	+	[2,25,26,27,31,53,56,65,67,70,72,78,88,90]
		−	No studies
		NS	[6,30,54,55,71,73,75] [58] (specific age range not stated) [53] (males, <40 years)
	Early middle age (30–45 years above)	+	[2,27,70,90] [53] (significant for all patients <40 years (*p* = 0.034), yet in terms of sex only females significantly associated <40 years (*p* = 0.002))
		−	No studies
		NS	[40,55,75] [53] (not significant in males <40 years) [58] (specific age range not stated)
	Late middle age (46–60 years above)	+	[2,90]
		−	No studies
		NS	[55,75] [58] (specific age range not stated)
	Older age (61 years+)	+	[28] (older patients had less recall, rating themselves non-adherent)
		−	[54,55]
		NS	[58,75] (specific age range not stated)
	Increased feeling of being between adolescence and adulthood	+	[57]
		−	No studies
		NS	No studies
Age at Diagnosis	Younger age at diagnosis (Up to 29 years)	+	[74,84]
		−	No studies
		NS	[25,72]
Sex	Female	+	[2,11,62,90] [59] (in UC) [83] (in whole population and CD, but not UC) [53] (in terms of sex, only females <40 years old)
		−	No studies
		NS	[27,30,75]
	Male	+	[53,67]
		−	No studies
		NS	No studies
Race	African–Caribbean descent	+	No studies
		−	No studies
		NS	[25]
Diagnosis	Crohn’s disease	+	[26,54,60,89]
		−	No studies
		NS	[6,8,33,75,93]
	Ulcerative colitis	+	[11]
		−	No studies
		NS	[8,75,93]
	IBD unclassified	+	No studies
		−	No studies
		NS	[8,26]
	Distal involvement (ulcerative colitis)	+	[59]
		−	No studies
		NS	No studies
	Perianal/perineal disease	+	[93]
		−	No studies
		NS	No studies
	Length of time since diagnosis	+	[57] (increased time since diagnosis) [78] (“short” diagnosis duration, ≤5 years) [32] (“long” diagnosis duration, 6–15 years) [83] (“long” diagnosis duration, 6–15 years with whole population and UC, but not CD)
		−	[33] (“long” diagnosis duration, 6–15 years)
		NS	[75] (“short” diagnosis duration, ≤5 years) [54,55,75] (“long” diagnosis duration, 6–15 years)
	Disease activity	+	[74,86,93] (active disease/not in remission); [66] (lower relapse probability) [57] (inactive disease/in remission) [44] (absence of physical bleeding)
		−	[33] (having at least one relapse in past 12 months)
		NS	[25,26,72] (active disease/not in remission) [11,25,66] (inactive disease/in remission) [74] (being in pain/discomfort)
Treatment	Aminosalicylates	+	[11,30,31,70,71,78]
		−	No studies
		NS	[8]
	Thiopurines	+	[44] (no concomitant use of thiopurines)
		−	No studies
		NS	No studies
	Biologics	+	[73] (intravenous/self-administering biologics) [75] (greatest non-adherence in combination therapy of biologic and immunomodulator, then infliximab, then adalimumab) [87] (intravenous/self-administering biologics only within 1st 12 months of treatment) [6] (intravenous/self-administering biologics; greatest non-adherence in adalimumab (43%), then infliximab (8%)) [88] (self-administering biologics only) [57] (not using biologics)
		−	No studies
		NS	[8] (self-administering biologics only) [89] (intravenous and self-administering biologics) [11] (not using biologics)
	Steroids	+	[86] (prescribed steroid script) [2] (not using steroids)
		−	[8] (prescribed steroid script)
		NS	[32]
	Antibiotics or topical steroids	+	[31] (either)
		−	No studies
		NS	No studies
	Immunosuppressants	+	[3] (not using immunosuppressants)
		−	No studies
		NS	[32]
	Biologics/immunosuppressants	+	[90] (no use of biologics/immunosuppressants within 12 months post-index date)
		−	No studies
		NS	No studies
	Dose	+	[66,79,82] (frequent/multiple-daily dose, e.g., ≥3 times daily) [33] (regimen of 40 mg adalimumab every other week as opposed to 40 mg every week) [90] (starting multiple daily dosing of either: balsalazide, mesalamine-delayed release (Asacol) or sulfasalazine) [44] (less frequent/fewer daily medications; <8 daily tablets)
		−	[33] (frequent/multiple-daily dose; regimen of 80 mg every other week, as opposed to 40 mg every other week)
		NS	[75] (less frequent/fewer daily medications—monotherapy) [90] (starting once-daily regime: “Multi-Matrix System” mesalamine/Lialda)
	Ongoing/lengthy treatment	+	[62,82]
		−	No studies
		NS	No studies
	Pill-burden (e.g., problem with dosing time regimen, high pill frequency or pill size)	+	[47] (time of dosing or pill size)
		−	No studies
		NS	[30]
	Route	+	[28] (subcutaneous rather than oral) [90] (not using rectal 5-ASA) [78] (topical medication)
		−	No studies
		NS	[32] (topical medication)
	Presence of adverse/side effects	+	[30,56]
		−	No studies
		NS	[66]
	Induction treatment	+	[90] (no history of switching from induction medication) [6] (anti-TNF induction)
		−	No studies
		NS	No studies
Healthcare	Care perspectives	+	[9,11,30,72] (negative relations/poor communication with HCP)
		−	[29] (perception of easy contact with gastroenterologist)
		NS	[29,40] (negative relations/poor communication with HCP) [8,52] (lack of trust in gastroenterologist) [29,30,40] (poor patient satisfaction)
	Care experienced	+	[66] (lack of treatment information from clinical team) [27,40,90] (no specialist/tailored care/follow-up by GP) [66] (lack of physician reinforcement regarding importance of treatment adherence) [65] (≤1 month between outpatient clinic appointment) [7] (frequent emergency care) [7,86] (frequent inpatient hospitalisation) [83] (no history of IBD related surgery, CD patients) [7] (fewer all-cause healthcare appointments)
		−	No studies
		NS	[9] (lack of/poor treatment information from clinical team) [9] (lack of involvement in prescribing) [8] (frequent inpatient hospitalisation) [29,31,93] (no history of IBD related surgery)
	Medication-taking behaviour close to timing of clinic visits	+	[82]
		−	No studies
		NS	No studies
General Health	Receiving treatment for other chronic condition	+	[66] (use of treatment for other chronic condition/s) [29] (use of treatment for other chronic condition/s when prescribed 5-ASA for IBD); [84] (not prescribed other chronic treatment)
		−	No studies
		NS	[55] (use of treatment for other chronic condition/s)
	Having a disability certificate	+	No studies
		−	No studies
		NS	[29]
	Comorbidities	+	[66]
		−	No studies
		NS	No studies
Habits	Smoking	+	[3,29,54] (current smoker) [53] (male only current smoker)
		−	[67] (current smoker) [54] (non-smoker)
		NS	[73,74,88,93] (current smoker) [89] (current smokers with CD)
	Alcohol consumption	+	[53] (whole population)
		−	No studies
		NS	[58] [53] (females only)
	Prescribed narcotic use (at time of biologic initiation)	+	No studies
		−	No studies
		NS	[89]
Diet	Frequently eating alone	+	[71]
		−	No studies
		NS	No studies
	Frequently missing a meal	+	[79]
		−	No studies
		NS	No studies
	Not storing treatment near to where meals are eaten	+	[42]
		−	No studies
		NS	No studies
	Use of nutritional supplements	+	No studies
		−	[31]
		NS	No studies
Finance	Healthcare/pharmacy prescription costs	+	[7] (lower UC pharmacy prescription patient costs and/or overall higher healthcare costs for patients)
		−	No studies
		NS	[66] (higher healthcare cost perception for patient appointments/treatment)
	Income/socioeconomic status	+	[25] (lower socioeconomic status)
		−	[41] (higher income)
		NS	[32] (higher socioeconomic status)
	Having public/non-commercial insurance	+	[88,89,90]
		−	No studies
		NS	No studies
Living Location	Country of residence (UK instead of Australia)	+	No studies
		−	[8]
		NS	No studies
	Living in north-east, south or west America	+	[90]
		−	No studies
		NS	No studies
	Living in mid-west America	+	No studies
		−	No studies
		NS	[90]
	Poor residential area/Poor QoL	+	[8,63,66] (reduced/poor quality of life) [26,41] (living in a poor residential area)
		−	No studies
		NS	No studies
Employment	Employment type	+	[11] (employed)
		−	[70] (self-employed)
		NS	[30] (permanent employment)
	Professional constraints	+	[56] (demanding jobs) [73] (work rhythms/constraints relating to treatment)
		−	No studies
		NS	No studies
Education	Educational level	+	[59] (lower educational level) [58,70] (higher educational level)
		−	No studies
		NS	[32,78] (higher educational level)
	Combined higher educational, occupational and socioeconomic status	+	[32]
		−	No studies
		NS	No studies
Social support + Relationships	Being single	+	[70]
		−	No studies
		NS	[93]
	Poor/low social support	+	[30] (received, emotional, tangible) [57] (informational)
		−	No studies
		NS	[57] (received, emotional, tangible)
	Having friends	+	[59] (dealing with friends when experiencing CD)
		−	No studies
		NS	No studies
Psychology	Treatment beliefs/perceptions/concerns	+	[30,66] (belief treatment is ineffective) [74] (belief treatment is ineffective in CD) [8,56,69,71] (belief there is no/little need for treatment/scepticism) [40] (negative beliefs about medication and poor patient satisfaction) [72] (lower perceived competence with treatment regime) [11,74] (lower perceived control over disease) [44] (negative beliefs about taking aminosalicylates: susceptibility, severity, benefits, barriers and cues to action) [47,69] (side effect concerns) [58] (adverse effect concerns)
		−	[58] (belief there is a need for treatment)
		NS	[65,78] (belief treatment is ineffective) [44] (belief for no/little need for treatment/scepticism) [9] (lower perceived competence with treatment regime) [30] (lower perceived control over disease) [66] (side effects and efficacy concerns) [71] (potential for harm of medication in general concerns) [6,8] (potential adverse effects concerns)
	Illness beliefs	+	[6] (shorter timeline perception/perceptions of IBD as an acute episodic disease) [59] (shorter perceived illness duration in CD; “perceptions that CD will end too soon”) [74] (shorter perceived illness duration in UC) [6] (illness identity)
		−	No studies
		NS	[6] (illness consequences)
	Depressive symptoms/antidepressant use/ psychiatric history	+	[25,58,66,67,74,92] (depressive symptoms/antidepressant use) [30] (patient-reported diagnosis and/or depression score from HADS)
		−	No studies
		NS	[89] (history of psychiatric disease in CD) [88] (comorbid psychiatric disease with IBD, e.g., depression and/or anxiety)
	Mood and attitude	+	[74] (low influence of disease on mood) [56] (indifferent attitude/less bothered regarding treatment benefit) [6] (stronger emotional response/negative emotions resulting from IBD)
		−	No studies
		NS	[73] (low influence of disease on mood) [72] (feeling stressed) [11] (lower sense of coherence)
	Anxiety	+	[30,58,73,74]
		−	No studies
		NS	No studies
	Negative religious coping (questions, doubt and strain around sacred matters with the divine, oneself and others)	+	[46]
		−	No studies
		NS	No studies
Accessibility, Organisation and Planning	Forgetting/disorganisation	+	[40] (forgetfulness) [30,71] (missing scheduled appointments)
		−	No studies
		NS	[63] (forgetfulness/carelessness)
	At weekends	+	[82]
		−	No studies
		NS	No studies
	Not keeping medications accessible when due	+	[42]
		−	No studies
		NS	[67]
	Not being as careful when taking medications	+	[9]
		−	No studies
		NS	No studies
	Lower priority for medications	+	[47]
		−	No studies
		NS	No studies
	Fewer cues to action (e.g., reminders to take medication)	+	[44]
		−	No studies
		NS	No studies
	Not using adherence tools (e.g., dosette boxes, alarms)	+	[42]
		−	No studies
		NS	No studies
Knowledge and Understanding	Poor/inadequate disease/ treatment knowledge	+	[11] (poor understanding of IBD as a disease) [59] (poor understanding of specifically CD) [74] (poor understanding of specifically UC) [58,65] (poor treatment knowledge) [66] (having inadequate information about treatment)
		−	No studies
		NS	No studies
	Poor recall of medical information	+	[28]
		−	No studies
		NS	No studies
	Internet use	+	[41] (not keen on using internet)
		−	No studies
		NS	No studies
	Being an information seeker/ having high curiosity	+	[31] (being an information seeker)
		−	No studies
		NS	[30] (having high curiosity)
	Patient organisation membership	+	[73]
		−	[8]
		NS	No studies
Alternative treatments	Complementary and alternative medicine (CAM) use	+	[63]
		−	No studies
		NS	[75]

Abbreviations: GP: general practitioner (family doctor); HADS: Hospital Anxiety and Depression Scale; HCP: Healthcare professionals; QoL: quality of life; UK: United Kingdom. Key: + = positive association with non-adherence; − = negative association with non-adherence; NS: non-significant association with non-adherence on MVA/factor analysis; blank cells = data not reported; [70]: age younger than 40 years old.

**Table 3 pharmacy-13-00021-t003:** Design of qualitative studies.

Reference	Qualitative or Mixed Methods	Data Collection Methods
[36],	Qualitative	Focus group
[33]	Quantitative and qualitative	Questionnaire with free text options
[34]	Quantitative and qualitative	Interviews and focus groups
[37]	Qualitative	Interviews and focus groups
[35]	Qualitative section within largely quantitative paper	Data from IBD Spanish database: electronic medical records reviewed
[38]	Qualitative	Social media posts: online with descriptive content analysis conducted
[29]	Qualitative section within largely quantitative paper	Online questionnaire with free text options
[28]	Qualitative section within largely quantitative paper	Recorded consultation with nurse (questionnaire completion pre and post) and 3-week follow-up telephone interview
[30]	Qualitative section within largely quantitative paper	Self-administered questionnaires with free text responses
[31]	Qualitative section within largely quantitative paper	Self-administered questionnaire with free text responses
[32]	Qualitative section within largely quantitative paper	Interviews with questionnaire followed by free text responses
[9]	Qualitative section within largely quantitative paper	Two open-ended questions within a questionnaire. Questionnaire completion via email/during clinic visits and medical records reviewed
[27]	Qualitative section within largely quantitative paper	Multiple choice questionnaire with some open-ended questions

**Table 4 pharmacy-13-00021-t004:** Qualitative data/free-text analysis/specific reasons reported for non-adherence.

Theme	Reasons for Non-Adherence	Studies
Disease/condition	Feeling better/being in remission	[29,32]
	Feeling unwell/hospitalisation	[33]
	No effect of medication/worsening of disease	[32]
Treatment	Side effects/adverse effects	[29,30,32,33,34,36,37,38]
	Complicated/difficult administration mode (pill size/discomfort/pain)	[30,32,34,36,37]
	Too many drugs/frequent drug dosing/regimen	[32,37]
	Pill fatigue	[37]
	Life-long treatment	[32]
	Treatment response time	[37]
	Drug safety	[38]
Healthcare	Distrust/poor confidence in healthcare provider	[38]
	Lack of convincing benefit based on doctor’s explanation	[30]
Background and general health	Having an infection	[33,36]
Eating	Fasting	[32]
Finance	Treatment cost	[29,30,32,36,37]
Work/occupation	Not taking treatment to work	[32]
Lifestyle	Busy life	[34,37]
	Change of routine (weekend/vacation)	[37]
	Being in public/social stigma	[37]
	Travel/away from home	[32,33,36,37]
Beliefs	Perception of treatment necessity	[29,37]
	Treatment fear, anxiety and uncertainty	[38]
	Stress/pressure	[38]
	Scepticism about treatment efficacy	[30,37]
	Treatment being disease reminder	[36]
	Disease non-acceptance	[36]
	Intentional non-adherence	[30,33,34]
Forgetting and organisation	Forgetting	[29,30,32,33,36,37]
	Timing/carelessness/disorganised	[30,37]
Accessibility	Treatment accessibility (including through GP/pharmacies/hospitals)	[29,32,33,36]
	Running out of treatment (whilst at home)	[29,32]
	Refill inconvenience	[37]
Knowledge and understanding	Lack of understanding regarding treatment regime	[32]
	Lack of understanding regarding treatment benefits	[37]
“Alternatives” to prescribed treatment	Using “healthier” alternatives	[38]
Pregnancy and pregnancy planning	Infertility	[38] *
	Pregnancy/avoiding perceived harm for current baby	[33,38]
	Avoiding all treatment for their next pregnancy	[38]
Independent research	Information gathered from online sources/online communities	[38]
Non-disclosed	Personal reasons	[38]

Abbreviations: GP: general practitioner/family Doctor. Key: * [38]: Infertility concerns were reported by both male and female participants with regards to taking IBD medications. Note: Ref. [35] collected and analysed qualitative variables, recording these according to the international classification of diseases (ICD), including chronic and psychiatric pathologies, expressed as frequencies (%). However, no demographic, phenotypic factors of the disease or therapeutic regimes were predictors of thiopurine non-adherence. Ref. [64] collected qualitative data regarding information-seeking sources and themes, but not the reasons for non-adherence, and thus not data relevant for the above table.

**Table 5 pharmacy-13-00021-t005:** Themes, sub-themes and qualitative quotations for non-adherence.

Theme	Sub-Themes for Non-Adherence	Qualitative Quotations for Non-Adherence
Treatment	Side effects/adverse effects	“Because of fears of side effects.” [29]
		“I’ve had some that make me jerk like a puppet…Side effects that you didn’t know, didn’t need and don’t want, and it’s so bad for you; you just stop because it’s… too much.” [36]
	Drug safety	“Humira is so new that most Dr’s [doctors] don’t have a clue when we ask about complications.” [38]
	Complicated/difficult administration mode (pill size/discomfort/pain)	“Those ones in the leg, would just, aaaarghh (shudders), and I know it’s coming and it was really hard… I felt sore… taking it all the time.” [36]
	Pill fatigue	“That’s really the biggest thing… I just have to take it in the morning, and then every once in a while, I’m just sick of taking it.” [37]
	Too many drugs/frequent drug dosing/regimen	“I take four of one kind twice a day; it would be awesome if that could be reduced down to one pill… ‘cos by the time you’ve had three devils… you choke on the pill… the big horse ones which have a nice coating, but they still get stuck down your throat.” [36]
Finance	Treatment cost	“Because medication is expensive.” [29]
		“If you get a repeat prescription and you had to go to the GP and they said so that’s gonna be $60 or a $100, I would go: I don’t have $100 or I have $100 but it makes a kind a financial thing.” [36]
Lifestyle	Travel/away from home	“Boxes of medications that’s just especially if you’re travelling overseas… makes for a very bulky parcel, and then there’s sometimes you get to your hotel room and you don’t have a fridge… it’s pretty much a nightmare, pretty challenging.” [36]
Beliefs	Disease non-acceptance	“So you develop an intense dislike that you have to take them because it makes you angry.” [36]
Forgetting and Organisation	Forgetting	“Because I forget.” [29]
	Timing/carelessness/disorganised	“I have a terrible memory so may have forgotten and just not realised… I take it a couple hours later when I remember.” [37]
Accessibility	Treatment accessibility (including through GP/pharmacies/hospitals)	“Because medication is not available in pharmacies.” [29]
	Running out of treatment (whilst at home)	“Because I run out of medications before I get a new prescription.” [29]
	Refill inconvenience	“When you go to refill it and you’ve passed the pharmacy hours or something. You just forgot or it wasn’t convenient.” [37]
		“The week before when you pick up your last repeat, you’ve got to then email. And sometimes they get it, sometimes they don’t… or it could be in their spam box.” [36]
Pregnancy and Pregnancy planning	Infertility	“My wife and I are most worried about having children soon or in the future but based on my research, you should not try while on the medication. Does anyone know any info on this? Please help!” [38]
	Pregnancy/avoiding perceived harm for current baby	“My doctor now wants me to take Asacol HD and I’m very hesitant to take any medication while pregnant for fear it may cause some kind of issue or birth defect with my baby.” [38]
		“Humira has not been fully studied in pregnant women… I know of a horror story and pregnancy and humira.” [38]

## Data Availability

Data supporting this study are included within the article and/or Appendix A.

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
