# Peer review of "Medication Non-Adherence in Inflammatory Bowel Disease: A Systematic Review Identifying Risk Factors and Opportunities for Intervention"

_pharmacy, 2025, doi:10.3390/pharmacy13010021_

Round 1
Reviewer 1 Report
Comments and Suggestions for Authors
An inspiring review that contributes to a better understanding of non-adherence in IBD. The authors have defined the inclusion criteria rather broadly in order to describe as many influencing factors as possible. This is a good approach, even if it makes the evaluation more difficult.
The distinction between intentional and unintentional non-adherence, modifiable and not modifiable factors is relevant. Some minor comments:
Introduction
The introduction is brief. Some statements should be confirmed with references:
Line 58/9: ‘Non-adherence in IBD leads to high healthcare and societal costs.’ [Kane & Shane 2008, doi 10.1007/s10620-007-9968-0] or [Park 2020, doi 10.1093/ibd104].
Line 63-5: ‘Several theories have been proposed as to why people are non-adherent to their medication, with recognition that some factors can be modifiable.’ Please tell us what these interesting theories say and provide the references.
Line 57: check the brackets
Results
The problem of rectal administration deserves special mention in the introduction and/or discussion section [ECCO guidelines 2022, doi 10.1093/ecco-jcc/jjab178]. Also valuable [Boyle 2015, doi 10.1097/MIB.0000000000000562], [D’Inca 2008, doi 10.1111/j.1365-2036.2007.03555.x]
Line 366ff: Specification of probability: I know p-value with small p
Discussion
Line 664/5: Special routes of administration are discussed very briefly ‘(e.g., injections, rectal medications or oral tablets difficult/ large to swallow)’. Rectal treatment, for example, is a specific challenge which could be discussed in more detail.
The statement in line 682/3 ‘…require hospital appointments for administration, raising non-adherence risks[22]’ suggests a particularly high non-adherence for intravenous administration of biologics in hospital. Does Reference 22 support this statement? Passive administration in hospital differs considerably from active intake or independent use by patients.
The sentence ‘Forward and backward translation prior use is therefore recommended.' in line 793/4 may be removed.
Author Response
Thank you for your comments. Please see the attachment for our responses.

Reviewer 2 Report
Comments and Suggestions for Authors
The authors present a systematic review of studies of factors that associated with non-adherence to treatments for inflammatory bowel disease. There is no question that adherence to disease modifying treatments is critical in preventing relapses in a wide range of autoimmune diseases, including IBD.
A) My first question is what led them to do this review now? I ask this because there are published systematic reviews that appear relevant to this question. They are not cited by the authors. They fall into 2 groups.
a. Systematic reviews of the literature identified by the authors. For instance, a study in 2010 in the American Journal of Gastroenterology (Jackson et al Am J Gastroenterol 105(3):p 525-539, March 2010.) See also Peel et al Gastrointestinal nursing, 2015-11, Vol.13 (9), p.17-24. There are probably more – I didn’t do a detailed search. Both these papers come to similar conclusions to the present publication. My question is: in the light of previous published work what additional value did the authors think another review would provide? Was it because of new key publications, new interventions, a different methodological focus of their review? I think this needs to be made clear.
b. Additionally, there have been systematic reviews of factors associated with adherence to treatments for a range of autoimmune diseases, which specifically mention IBD. For instance, Vangeli et al Adv Ther (2015) 32:983–1028
B) My next question is whether the authors consulted the broader literature on factors associated with non-adherence with treatments for a broader range of chronic diseases? There is an umbrella reviews that summarises these and some findings are likely to be relevant to IBD (Mathes et al. Archives of Public Health 2014, 72:37.)
C) The review itself appears to be competent. The literature searches, definitions of variables and identification of candidate variables (associated with non-adherence) appear sound. The authors have used a study quality assessment instrument.
D) The existing literature on non-adherence with chronic disease therapies could have guided the authors to hypothesise that some variables were a priority for study. But I did not see evidence of this.
E) The review is badged as a systematic review but the statistical summary of results is by ‘head counting’ listing studies that did or did not find associations is fairly crude. It may be that the authors felt the available data didn’t enable anything more sophisticated. For instance in situations where some studies found an association, some found a non-significant trend while others were null, or even a trend in the opposite direction did they consider using meta-analysis or meta-regression or were data too sparse?
F) The layout of the Tables made reading the article difficult. Table 4 runs out to 7 pages and Table 4 to 3 pages, interrupting the flow of the results sections. I think that the authors should make these supplementary files and derive a summary table that pulls the information together succinctly. References in these tables can use numeric format to make it more compact.
G) The Discussion in my view needs work. It is very long – around 2900 words, close to the length of a full manuscript. The discussion of the internal results of this study is excessively detailed. I think the main points could be summarised in about half the length they have used. As noted above there is no comparison with previously published reviews. The first sentence reads ‘This is the first comprehensive, international review, that outlines the complexities and challenges of non-adherence to prescribed medication in IBD’. You can only make such a statement when you are sure that you have searched and evaluated previously published systematic, scoping and narrative reviews. I don’t think the authors have done that, or at least they don’t mention it. They should compare their own work with these reviews and show how it advances knowledge in this field.
Author Response

(The authors gave the same response as above.)

Reviewer 3 Report
Comments and Suggestions for Authors
pharmacy-3413027
"Factors associated with Medication Non-adherence in Inflammatory Bowel Disease Patients: A Systematic Review."
The manuscript reports an important and challenging topic in the use of medicines.
Check error types - e.g.
line 10 - Abstract: A single paragraph of about 200 words maximum.
Assessment of Title Accuracy
· Take care of the following suggestion
o The phrase "Factors associated with Medication Non-adherence" is broad and does not emphasize the dual focus on modifiable and non-modifiable factors.
o The title could be more engaging and precise by highlighting the implications or significance of the findings.
Improved Title Options
1. "Modifiable and Non-modifiable Factors Influencing Medication Non-adherence in Inflammatory Bowel Disease: A Systematic Review"
o Highlights the dual focus on modifiable and non-modifiable factors.
2. "Understanding Medication Non-adherence in Inflammatory Bowel Disease: A Systematic Review of Associated Factors"
o Emphasizes the study’s purpose of understanding the issue.
3. "Medication Non-adherence in Inflammatory Bowel Disease: Identifying Risk Factors and Opportunities for Intervention"
o Adds a forward-looking perspective by hinting at potential interventions.
Key questions to guide improvements in the manuscript
Abstract and Introduction
1. Abstract
· Can you clarify the most critical findings and their implications in the abstract? Consider condensing the range of non-adherence percentages to avoid confusion.
· How do the findings specifically contribute to addressing gaps in current IBD management practices?
2. Introduction
· Could you expand on the significance of studying modifiable vs. non-modifiable factors in medication adherence?
· Can you provide more context on how this review differs from or builds upon previous systematic reviews in this area?
Methodology
3. Search Strategy and Inclusion Criteria
· Why were studies before 2011 excluded? Could older studies provide valuable historical context or trends?
· Were any steps taken to assess the potential for publication bias in the included studies?
4. Data Analysis
· Can you elaborate on why multivariable analysis (MVA) was emphasized over other methods? Were there limitations in studies that did not use MVA?
· How did you address the variability in adherence definitions across studies to ensure meaningful comparisons?
Results
5. Presentation of Data:
· The range of non-adherence (4.3%–88.9%) is very broad. Can you provide more context or subgroup analyses to explain this variability?
· Could you clarify which factors (modifiable or non-modifiable) had the strongest and most consistent associations with non-adherence?
6. Demographics and Subgroup Analysis:
· Were there significant differences in non-adherence based on specific demographics (e.g., age, sex, geographic location)? If so, how do these findings inform targeted interventions?
· Were there any unique findings in specific subgroups, such as adolescents or pregnant women, that warrant further discussion?
Discussion
7. Contradictory Findings
· How do you explain the contradictory results across studies for factors like age, sex, and disease type?
· Can you provide more concrete recommendations for clinicians based on the findings, especially for addressing modifiable factors?
8. Practical Implications
· How can the identified factors be integrated into clinical practice to improve adherence?
· Are there examples of successful interventions from the included studies that could be highlighted?
Conclusion
9. Actionable Insights
· Could you refine the conclusion to emphasize actionable insights and specific areas for future research?
· What are the most urgent gaps in knowledge that should be addressed in subsequent studies?
Figures and Tables
10. Data Visualization
· Some tables and figures are dense and difficult to interpret. Can you simplify or summarize key data points for clarity?
· Could you add visual aids, such as graphs or infographics, to better represent trends or associations?
General
11. Language and Clarity
· Certain sections, especially the results and discussion, are highly technical. Can you simplify the language for a broader audience, including clinicians and policymakers?
· Are there specific terms or definitions (e.g., adherence, non-adherence) that need to be standardized throughout the manuscript?
12. Limitations
· Could you expand on the limitations of the included studies and the review itself? For example, were there challenges in synthesizing data due to differences in study designs or definitions?
Recommendations for Improvement
1. Abstract and Introduction
· Topic The abstract provides an overview but lacks clarity in presenting the key findings and their implications. The introduction, while informative, could better frame the study's significance in the broader context of healthcare.
· Recommendation Revise the abstract to highlight the study's main findings and their clinical relevance more succinctly. In the introduction, emphasize the study's contribution to addressing gaps in understanding medication adherence in IBD patients.
2. Methodology
· Topic The methodology section is thorough but includes redundant details, such as overly extensive descriptions of search terms and databases.
· Recommendation Streamline the methodology to focus on critical aspects like inclusion criteria, data extraction processes, and analytical methods. Provide additional justification for using specific analytical tools, such as MVA, and explain how biases were minimized.
3. Results
· Topic The results section presents a vast amount of data but lacks synthesis. The broad range of non-adherence rates (4.3%-88.9%) is not well contextualized, which may confuse readers.
· Recommendation Condense the presentation of data by focusing on the most impactful findings. Provide a narrative that ties together the statistical results and their practical implications.
4. Discussion
· Topic The discussion does not adequately address the contradictions in the findings or propose solutions for clinical practice based on the study results.
· Recommendation Expand the discussion to explore why certain factors (e.g., younger age, female sex) are associated with non-adherence in some studies but not others. Offer practical recommendations for clinicians to address modifiable factors.
5. Figures and Tables
· Topic Some tables and figures, such as those summarizing non-adherence rates and factors, are overly detailed and difficult to interpret at a glance.
· Recommendation Simplify tables and figures to focus on key data points. Use visual aids like graphs or summary charts to enhance readability.
6. Language and Formatting
· Topic The manuscript contains technical jargon and complex sentence structures that may hinder comprehension.
· Recommendation Use simpler language and shorter sentences to improve accessibility. Ensure formatting adheres to the journal's guidelines.
7. Conclusion
· Topic The conclusion reiterates the findings but does not emphasize actionable insights or future research directions.
· Recommendation Refocus the conclusion to highlight the practical implications of the findings and suggest specific areas for further investigation.
Overstated claims
1. Abstract
· Claim "Identifying factors associated with IBD medication adherence that can be targeted, is important to keep IBD quiescent."
· Overstated reason The phrase "important to keep IBD quiescent" implies that identifying these factors alone will ensure disease remission, which oversimplifies the complexity of IBD management.
· Suggested rephrasing "Identifying factors associated with IBD medication adherence is crucial for supporting effective disease management and maintaining remission."
2. Results
· Claim "Non-adherence to IBD medications limits drug effectiveness, being associated with a 5-fold greater risk of disease flare or worsening of the disease, disability, colorectal cancer, hospitalisation, therapy escalation and surgery, reduced quality of life and increased morbidity and mortality."
· Overstated reason While these outcomes are possible, the statement implies a direct and universal causal relationship between non-adherence and all these severe outcomes without acknowledging variability or other contributing factors.
· Suggested rephrasing "Non-adherence to IBD medications can significantly impact treatment outcomes, with studies associating it with increased risks of disease flare, hospitalizations, and reduced quality of life."
3. Discussion
· Claim "Treatment adherence is essential to maintain remission."
· Overstated reason While adherence is important, it is not the sole factor in maintaining remission, as other clinical and biological factors also play a role.
· Suggested rephrasing "Treatment adherence is a critical component in maintaining remission, alongside other clinical and biological factors."
4. Conclusion
· Claim "Clinicians aware of non-modifiable factors can identify relevant patients and support their adherence."
· Overstated reason This statement oversimplifies the complexity of clinical practice and implies that awareness alone is sufficient to improve adherence.
· Suggested rephrasing "Clinicians who are aware of non-modifiable factors can better identify patients at risk of non-adherence and develop targeted strategies to support them."
5. Figures and Tables
· Claim The range of non-adherence (4.3%–88.9%) is presented without sufficient context, potentially overstating the variability.
· Overstated reason The wide range may result from differences in study populations, definitions, and methodologies, which are not adequately explained.
· Suggested rephrasing Provide a note or clarification in the figure/table legend: "The range of non-adherence (4.3%–88.9%) reflects variations in study populations, adherence definitions, and measurement methodologies."
Author Response

(The authors gave the same response as above.)

Reviewer 4 Report
Comments and Suggestions for Authors
You performed a systematic review with the title "Factors associated with Medication Non-adherence in Inflammatory Bowel Disease", in which a total of 7,596 papers were identified from six databases which, after being analyzed, resulted in 79 analyzed.
The approach complies with the rules of systematic reviews, using the Check List provided by the BMJ
The work respects the appropriate formatting and is clear,
The aspects analyzed are extensive and throughly discussed
Limitations are openly expressed
The study is relevant and the conclusions are significant
ceania (3 studies)
Author Response
Thank you very much for taking the time to review this manuscript. We appreciate the positive reviews and comments and are very pleased to hear that you are happy with all areas of the Review.

Round 2
Reviewer 3 Report
Comments and Suggestions for Authors
Thank you for your feedback on the topics proposed in the review. Congrats on your paper.